# Pretraining boosts out-of-domain robustness for pose estimation

## Abstract

Deep neural networks are highly effective tools for human and animal pose estimation. However, robustness to out-of-domain data remains a challenge. Here, we probe the transfer and generalization ability for pose estimation with two architecture classes (MobileNetV2s and ResNets) pretrained on ImageNet. We generated a novel dataset of 30 horses that allowed for both within-domain and out-of-domain (unseen horse) testing. We find that pretraining on ImageNet strongly improves out-of-domain performance. Moreover, we show that for both pretrained and networks trained from scratch, better ImageNet-performing architectures perform better for pose estimation, with a substantial improvement on out-of-domain data when pretrained. Collectively, our results demonstrate that transfer learning is particularly beneficial for out-of-domain robustness.

## 1 Introduction

Pose estimation is an important tool for understanding behavior, as it belies analysis of movement kinematics, action recognition, and ethology (Mason & Lott, 1976; Heinroth & Heinroth, 1966; Ijspeert, 2014; Anderson & Perona, 2014). Pose estimation on humans has reached remarkable capabilities due to innovations in both algorithms (Insafutdinov et al., 2017; Cao et al., 2017; He et al., 2017; Alp Güler et al., 2018; Xiao et al., 2018; Kreiss et al., 2019; Sun et al., 2019) and large-scale datasets (Lin et al., 2014; Andriluka et al., 2014; 2018). However, it is a challenging problem due to small joints, occlusions, clothing, and changes in background and scene statistics. Thus, many networks suffer when applied to out-of-domain data, i.e. images that are sufficiently different from the training set. For instance, they fail on very articulated human movements like skiing, or other 'rare poses', if not in the training set (Rhodin et al., 2018; Dang et al., 2019). Moreover, animal pose estimation has additional challenges. Not all animals share the same keypoints, therefore a universal "animal pose detector" is not feasible. Even building animal-specific networks would require a lot of data, due to the large variability in body shapes, colors, and the number of species as well as breeds of a type of animal. Therefore, the question of how one can robustly learn from limited annotated datasets is of particular importance, and animal pose estimation datasets allow for generalization to be systematically tested (Novotny et al., 2017).

How can robustness be achieved? Transfer learning, or the transferability of pretrained features from one task to another, is a powerful approach that has been well studied in computer vision (Donahue et al., 2014; Yosinski et al., 2014; Kümmerer et al., 2016; Huh et al., 2016; He et al., 2018). It has been shown to improve performance on some human pose estimation tasks (Mehta et al., 2016; Mueller et al., 2017; Xiao et al., 2018; Insafutdinov et al., 2017), yet is not universally used in the top-performing networks on the human 2D/3D pose estimation benchmarks (Doersch & Zisserman, 2019). For keypoint detection, He et al. recently showed that pretraining on ImageNet did not result in overall performance improvements if randomly initialized models were allowed to train for much longer than usual, therefore suggesting that (given enough task-data) the main benefit of transfer learning is shorter training time, rather than performance (He et al., 2018). However, it has not been tested whether pretraining on ImageNet offers advantages in robustness, for instance as measured by any performance advantage on out-of-domain data.

We address this by building a new pose estimation task of $8,114$ labeled video frames from 30 Thoroughbred horses. We focus on horses as their diversity readily allows us to assess out-of-domain generalization, i.e. the ability to generalize to the different, unseen horses in different contexts. We

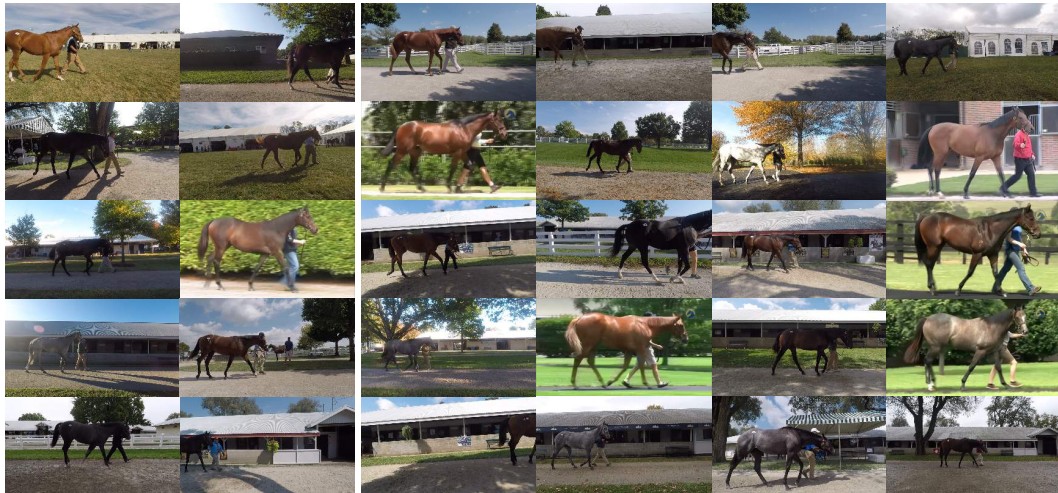

Figure 1: **Horse Dataset:** Example frames for each young Thoroughbred horse in the dataset. In each video the horse walks from left to right. The videos vary in horse color, the appearance of sunlight and shadow, and relative horse size as well as background. This makes the data set ideal for tests in robustness and generalization. To illustrate the horse-10 task we arranged the horses according to one split: the ten leftmost horses were used for train/test within-domain, and the rest are the out-of-domain held out horses.

created a task, called Horse-10, that uses only 10 horses for the test/train splits, and uses the other 20 horses to test out-of-domain performance (Figure 1). The data will be made available at TBA.

Here we report two key insights: (1) higher ImageNet performance leads to better generalization for both within domain and on out-of-domain data for pose estimation but with a stronger effect on out-of-domain data (see Figure 2B); (2) transfer learning improves robustness again most strongly for out-of-domain data, and yields up to 3 times more accurate results than training from scratch (see Figure 4D,E). Thus, while it has been previously shown that training from scratch can match performance on in-domain data for sufficiently large amount of training data and training times (He et al., 2018), we show it clearly cannot match performance of pretrained networks on out-of-domain data (see Figure 5). Collectively, this sheds a new light on the inductive biases of "better ImageNet architectures" for visual tasks to be particularly beneficial for robustness, even beyond within domain data accuracy, on out-of-domain datasets.

## 2 RESULTS

To test within and out-of-domain performance we created a new dataset of 30 different walking horses (Thoroughbreds that are led by different humans), resulting in a dataset of $8,114$ images with 22 labeled body parts each. Horses have various coat colors and the "in-the-wild" aspect of the collected data at various Thoroughbred yearling sales and farms added additional complexity. The sunlight variation between each video added to the complexity of the learning challenge, as well as the handlers often wearing horse-leg-colored clothing. Some horses were in direct sunlight while others had the light behind them, and others were walking into and out of shadows, which was particularly problematic with a dataset dominated by dark colored coats (Figure 1). Thus, this dataset is ideal for testing robustness and out-of-sample generalization.

### 2.1 IMAGENET ACCURACY PREDICTS ANIMAL POSE ESTIMATION ACCURACY

To probe the role of different ImageNet pretrained architectures, we compared four variants of MobileNetV2 with varying expansion ratios, as the width multiplier parameterizes ImageNet performance over a wide range (see Methods), and two variants of ResNets (50 and 101 layers deep). We utilized a 'simple' yet competitive pose estimation architecture (Insafutdinov et al., 2016; Mathis

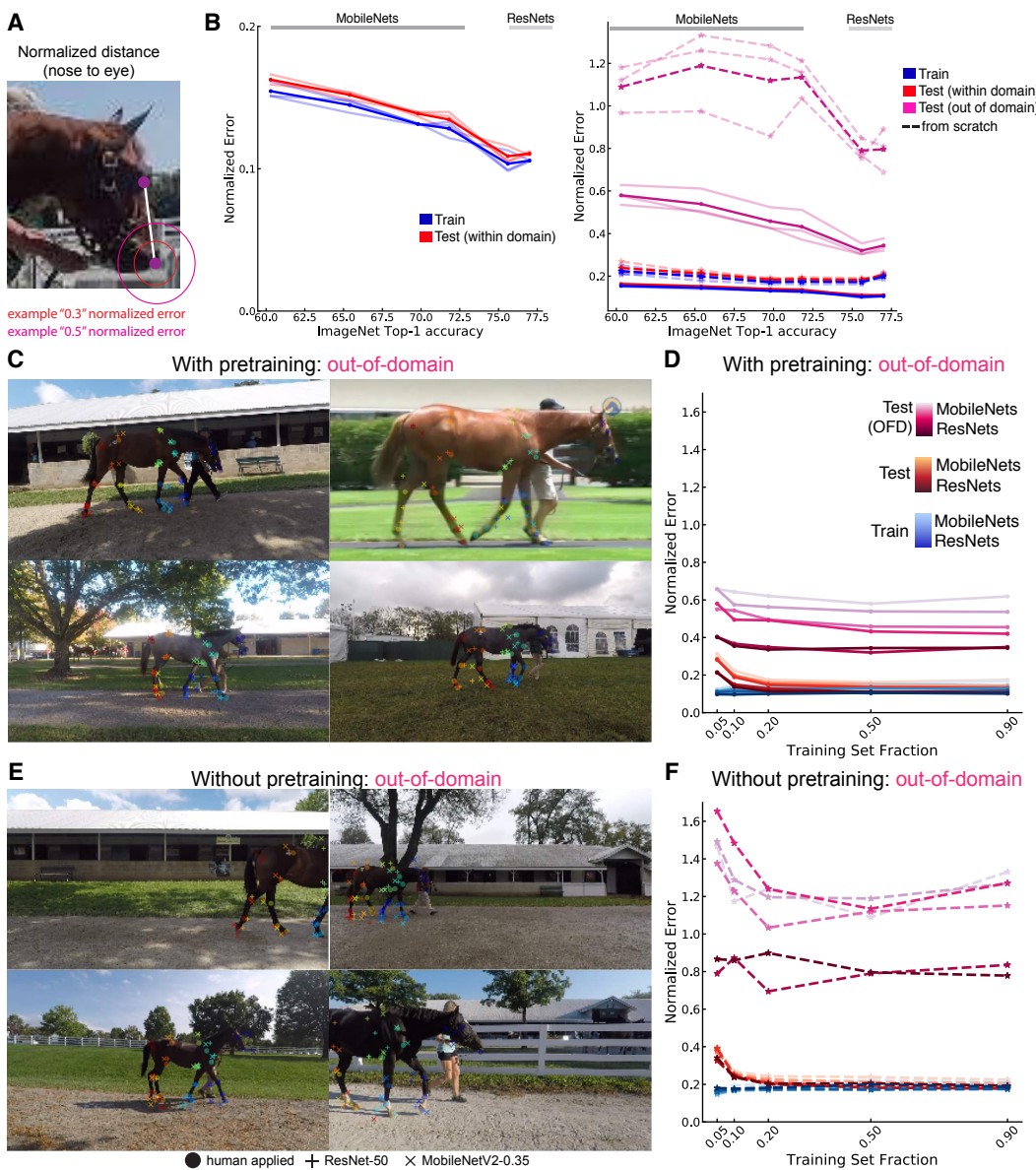

Figure 2: **Transfer Learning boosts performance, especially on out-of-domain data. A:** Illustration of the normalized error metric. **B:** Normalized Error vs. Network performance as ranked by the Top 1% accuracy on ImageNet (order by increasing ImageNet performance: MobileNetV2-0.35, MobileNetV2-0.5, MobileNetV2-0.75, MobileNetV2-1, ResNet-50, ResNet-101). The pose estimation performance is for $50\%$ training set fraction. The faint lines indicate data for the three splits. LEFT: Test data is in red, train is blue. RIGHT: additionally, pink is out-of-domain data; dashed lines indicate networks trained from scratch. Better ImageNet networks perform better on Horse-10; this relationship is even stronger for out-of-domain data. **C:** Example frames with human annotated body parts vs. predicted body parts for MobileNetV2-0.35 and ResNet-50 architectures with ImageNet pretraining on out-of-domain horses. **D:** Normalized Error vs. Training Set Fraction of Horse-10. For reference, 5% training data is $\approx 160$ frames. Darker to light red shades are test results for pretrained networks on within-domain data. Shades of pink show the test on out-of-domain data (order according to ImageNet performance: ResNet-101, ResNet-50, MobileNetV2-1, MobileNetV2-0.75, MobileNetV2-0.5, MobileNetV2-0.35). **E:** Same as C but for training from scratch. **F:** Same as D but for training from scratch. All lines are averages of 3 splits (see Methods).

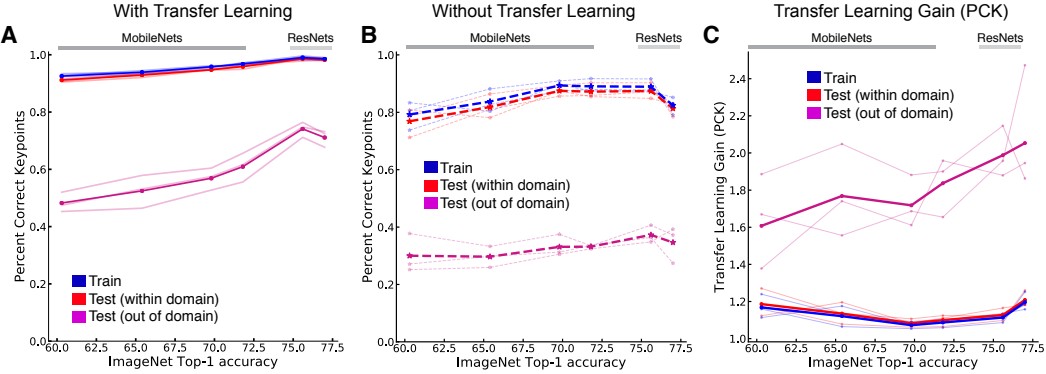

Figure 3: **Fraction of correctly identified bodyparts improves with transfer learning A:** Percent Correct Keypoint (PCK) vs. Training Set Fraction shows high performance for all pretrained networks on Horse-10. **B:** Same as A, but training from scratch. The performance drops strongly, especially for out-of-domain data **C:** Performance gain when using transfer learning. All lines are averages of 3 splits, individual splits are shown as faint lines.

et al., 2018) embedded in DeepLabCut, a toolbox for data-set generation, training, and evaluation (see Methods). The architectures then consisted of either MobileNetV2s (Sandler et al., 2018) or ResNets (He et al., 2016), where a single deconvolution layer is connected to the final convolutional layer to predict poses via body-part specific scoremaps as well as location refinement maps (Insafutdinov et al., 2016; Mathis et al., 2018). We created 3 splits containing 10 random horses each, and then varied the amount of training data from these 10 horses (referred to as Horse-10, see Methods). As the horses could vary dramatically in size across frames, due to the "in-the-wild" variation in distance from the camera, we used a normalized pixel error; i.e. we normalized the raw pixel errors by the eye-to-nose distance and report the fraction within this distance (Figure 2A). In total, we found that all pretrained-ImageNet networks showed great performance on Horse-10 within domain (Figures 2B, 7).

To further assess the errors we computed the percent correct keypoints (PCK; defined as within 30% of the distance from nose-to-eye, see Methods) and found that performance was nearly 97% for ResNets (with at least 20% training data) and only fell to $\approx$ 93% on MobileNetV2-based models (Figure 8A). Even with very small datasets (5%, i.e. around 160 training images) performance was 80% to 85% on MobileNetV2 and ResNets, respectively (Figure 8A).

Next, we directly compare the ImageNet performance to their respective performance on this pose estimation task. We find Top-1% accuracy on ImageNet, correlates with pose estimation error (linear fit: slope $-0.33\%$, $R^2 = 0.95$, $p = 0.001$; Figures 2B). This linear relationship is consistent with a recently reported correlation of ImageNet accuracy and performance for various object recognition tasks (Kornblith et al., 2019).

## 2.2 USING PRETRAINED-IMAGENET NETWORKS SIGNIFICANTLY BOOSTS OUT-OF-DOMAIN PERFORMANCE

The larger challenge is posed by the out-of-domain horses, rather than on different frames for same horses as used for training. Thus, we evaluated the performance of the networks that had been trained for various fractions of the training data and found that both MobileNetV2s and ResNets were robust (Figures 2B-D).

Most strikingly, on out-of-domain horses, the relationship between ImageNet performance and performance on Horse-10 was even stronger. This can be quantified by comparing the linear regression slope for out-of-domain test data: $-1.6\%$ pose-estimation improvement per percentage point of ImageNet performance, $R^2 = 0.95$, $p = 0.0008$ vs. within-domain test data $-0.33\%$, $R^2 = 0.95$, $p = 0.0010$ (Figures 2B-F). In other words, *less* powerful models (MobileNetV2s) seem to overfit more on the training data. We mused that this improved generalization could be a consequence of the

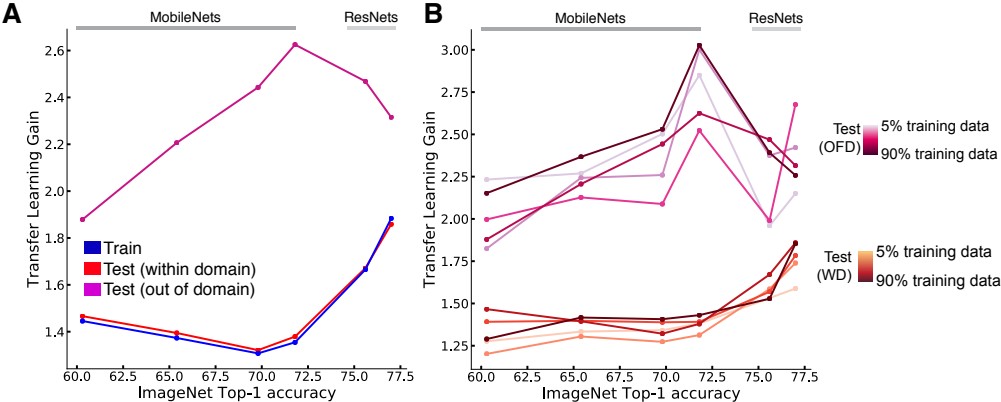

Figure 4: **Up to a 3X gain with transfer learning on out-of-domain data. A:** Transfer learning gain vs. architectures with $50\%$ of the data used for training (comparing pretrained networks to from-scratch from Figure 2B). **B:** Same as in A, but for varying levels of input data (5 to 90%), light to dark, respectively. All lines are averages of 3 splits.

ImageNet pretraining or the architectures themselves. Thus, we trained the different architectures only on the task itself.

### 2.3 TASK-BASED TRAINING FROM SCRATCH

To assess the impact of ImageNet pretraining we also trained all architectures from scratch. Thereby we could directly test if the increased slope for out-of-domain performance across networks was merely a result of more powerful network architectures.

When training from scratch directly on the task for the same amount of iterations (see Methods), we found that all networks performed well on within domain data, given enough training data. The ResNets once again showed an advantage over the MobileNetV2 variants. All the networks performed worse on within domain compared to pretrained-ImageNet networks, and strikingly $2X$ worse on out-of-domain data (Figures 2E,F).

The PCK for all networks as a fraction of the training set size also reflected this decrease in performance compared to pretraining (Figures 8A-C). For example, while PCK with pretrained ResNet network was nearly $97\%$ (with $20\%$ of the training data), without pretraining this falls to around $80\%$. Out-of-domain performance drops substantially (pretrained vs. randomized initial weights; comparing Figure 8B to Figure 8C).

Without pretraining we find that the Top-$1\%$ accuracy on ImageNet ranking of models only weakly correlates with pose estimation error (linear fit: slope $-0.26\%$, $R^2 = 0.53$, $p = 0.166$; Figures 2B and 9B). On out-of-domain horses the slope was similar (slope $-0.21\%$, $R^2 = 0.54$, $p = 0.098$), unlike when training from pretrained checkpoints. Taken together, our results suggests that ImageNet pretraining significantly boosts generalization (vs. just being a feature of the architectures themselves).

Next we quantified the amount of performance gain across all networks (with vs. without pretraining). We found an up to $2X$ gain in performance (increase in PCK) with transfer learning (Figure 3A-C). Remarkably, for both ResNets and MobileNetV2s, pretraining on ImageNet boosts within domain and out-of-domain reduction in pixel-errors (Figure 4A,B), with the largest gains on out-of-domain data - with $90\%$ of the training data there was a gain of up to a $3X$ (Figure 4B).

### 2.4 FROM SCRATCH NETWORKS CANNOT MATCH THE PERFORMANCE OF PRETRAINED-IMAGENET NETWORKS ON OUT-OF-DOMAIN DATA

He et al. (2018) recently showed that training ResNets directly on COCO data for object detection, instance segmentation and key point detection, catches-up with pretrained network accuracy when

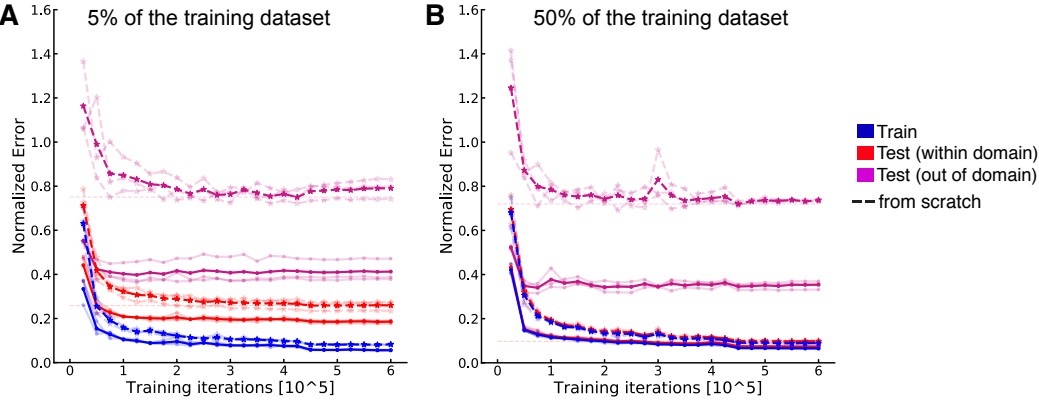

Figure 5: **Training randomly initialized networks longer cannot rescue out-of-domain performance. A:** Normalized error vs. training iterations for ResNet 50 using 5% of the training data. Training from scratch for $600,000$ iterations does not match the performance of pretrained condition after $100,000$ iterations. Out-of-domain testing does not approach pretrained levels of performance. Faint, dashed lines are backwards projecting from lowest from scratch performance to aid in visualization. **B:** Same as A but using 50% of the training data. Test errors when training from scratch closely match the transfer learning performance after many iterations. Crucially, out-of-domain testing does not approach performance for pretrained network.

training for $6X$ more iterations as typical training schedules. However, due to the nature of the task, they did not test this relationship on out-of-domain data. Given that we see the largest gains of transfer learning on out-of-domain data, we asked if the randomly initialized networks could also catch-up on that metric.

For all the network analysis so far, we trained for $100,000$ iterations (as the loss had relatively flattened), therefore we trained $6X$ longer to see if test/training errors would decrease. Indeed, consistent with He et al. 2018, we found that randomly initialized networks could closely match the performance of pretrained networks, given enough data and time (Figure 5A, B); for smaller datasets (5% training data), this was not the case (Figure 5A), again suggesting that pretrained-networks offer an advantage for small datasets, which is particularly important for applications in biology (Mathis et al., 2018).

Crucially, and most strikingly, for out-of-domain data this was not the case: the from-scratch trained networks never caught up (and indeed plateaued early; Figure 5A, B). Thus, transfer learning offers multiple advantages. Not only does pretraining networks on ImageNet allow for using smaller datasets and shorter training time, it also significantly improves robustness.

## 3 DISCUSSION

Here we report two key findings: (1) pretrained-ImageNet networks offer an advantage: shorter training times, less data requirements, and robustness on out-of-domain data, and (2) networks that have higher ImageNet performance lead to better generalization, especially on out-of-domain data. Recently, it was shown that for many object recognition datasets the transfer ability is improved when fine-tuning architectures with better ImageNet performance (Kornblith et al., 2019; Huh et al., 2016). In fact, Kornblith et al. (2019) find high correlation between between ImageNet and transfer accuracy for other recognition tasks ($r > 0.95$). In contrast to the exhaustive study by Kornblith et al., we only focused on two architecture types: ResNets (He et al., 2016) and MobileNetV2s (Sandler et al., 2018). However, we vary parameters of those networks to also span a broad range of ImageNet accuracies. Consistent with Kornblith et al, we find that ImageNet accuracy is weakly correlated with performance on pose estimation when trained from scratch ($R^2 = 0.53$), and strongly when fine-tuning ($R^2 = 0.95$). We also find that "better" ImageNet networks transfer better. Moreover, we show that transfer learning significantly improves performance on out-of-domain data.

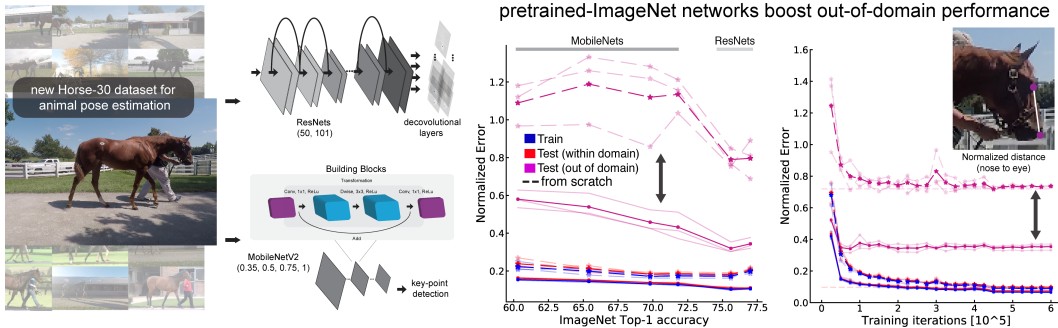

Figure 6: **Summary of Findings:** We present a new horse dataset for testing within and out-of-domain performance for pose estimation. We tested two classes of models, MobileNetV2s and ResNets, which span a wide range of performance on ImageNet. We find that networks that perform better on ImageNet are better for pose estimation. We also find that pretrained-ImageNet models strongly improve out-of-domain robustness.

### 3.1 TRANSFER LEARNING

We show that for both within and out-of-domain pose estimation tasks, transfer learning improves performance. Most notably, transfer learning boosts out-of-domain generalization, improving up to $3X$ compared to networks without pretraining, and even when training for much longer, as suggested in He et al. (2018), this gap cannot be closed.

Another important insight is that for small amounts of data, pretrained networks offer a large advantage (Figures 4 & 5). Corroborating He et al. (2018) we find that given enough training data, training from scratch, with purely task-driven training can match the performance of of transfer learning; however, we also found that for out-of-domain data, pretraining helps significantly, boosting performance up to 3 times (Figure 4).

### 3.2 ON THE IMPORTANCE OF DATA

Looking forward, this suggests that collecting annotations of task data (instead of pretraining data) is more useful. We think that benchmarks with different contexts, like Horse-10, are important to improve pose estimation algorithms for biological applications (i.e. for small-scale lab-based experiments). A future goal will be to limit, or remove, training altogether. However, in order to create networks that generalize across laboratories and setups, transfer learning will be important for robustness. Yet, more work needs to be done to close the gap between within domain and out-of-domain generalization.

What is the limit of transfer learning? Would ever larger data sets give better generalization? Interestingly, it appears to strongly depend on what task the network was pretrained on. Recent work by Mahajan et al. (2018) showed that pretraining for large-scale hashtag predictions on Instagram data (3.5 billion images) improves classification, while at the same time possibly harming localization performance for tasks like object detection, instance segmentation, and keypoint detection. This highlights the importance of the task, rather then the sheer size as a crucial factor. Further corroborating this insight, Li et al. showed that pretraining on large-scale object detection task can improve performance for tasks that require fine, spatial information like segmentation (Li et al., 2019). Thus, one interesting future direction to boost robustness could be to utilize networks pretrained on OpenImages, which contains bounding boxes for 15 million instances and close to 2 million images (Kuznetsova et al., 2018).

### 3.3 MOBILENETV2 FOR FAST POSE ESTIMATION

Here, we introduce using MobileNetV2 variants for animal pose estimation that can achieve high accuracy but with $2.5X$ the speed as a ResNet backbone (Figure 10), making pretrained-MobileNetV2 an excellent option for real-time applications in the wild (on mobile-phones) and in the laboratory.

If an end-user utilizes small training sets, ResNets offer an advantage, yet MobileNetV2s are significantly faster (Figure 10) and performs reasonably well, within domain; i.e. to match the ResNet-101 performance with $10\%$ of the training set one needs about $50\%$ for the best MobileNetV2. Potentially, the few pixels lost in accuracy is worth the significant speed improvement (twice as fast) for high-throughput experiments and for real-time applications. MobileNetV2 can run batch inference of ($> 2,500FPS$) on a GPU. Using MobileNetV2 also has other advantages: one, MobileNetV2 has low memory demands, and even runs on mobile phones, as the name suggests; two: on CPUs one gets even more speed improvements (Figure 10).

What are the trade-offs? With more data for training the MobileNetV2 match the performance of ResNets trained with less labeling data (Figure 2B). However, the ResNets still perform best with matched amounts of data. Thus, to close this gap "Student-Teacher networks" could be used. For example, one could build a larger and more robust ResNet-101 network, then run inference to generate a larger dataset to train the MobileNetV2 variant for fast inference on within domain data.

## 4 CONCLUSIONS

We found a significant advantage of using pretrained networks for out-of-domain robustness. While there is still a gap to close, we believe this work demonstrates that transfer learning approaches are powerful to build robust architectures. We also demonstrate that ImageNet performance correlates with animal pose estimation accuracy on a challenging "in-the-wild" new horse dataset (Figure 6). Moreover, we aim to add a new variant of networks to the open-source DeepLabCut project, MobileNetV2s, that pave the way for fast and accurate pose estimation. Collectively, our work highlights that pretrained networks require less training data, and allow for faster training, and boost robustness, especially for out-of-domain data.

## 5 METHODS

### 5.1 HORSE DATASET

Here we developed a novel horse data set comprising 30 different horses captured for $4-10$ seconds with a GoPro camera (Resolution: $1920 \times 1080$, Frame Rate: 60 FPS), which we call Horse-30. We used the DeepLabCut2.0 toolbox (Nath et al., 2019) for labeling. We downsampled the frames by a factor of $15\%$ to speed-up the benchmarking process ($288 \times 162$ pixels; one video was downsampled to $30\%$).

Using previously established anatomical landmarks for equine biomechanical evaluation (Magnusson & Thafvellin, 1990; Anderson & McIlwraith, 2004), the following 22 body parts were labeled by an expert in Thoroughbred horses [BR] across $8,114$ frames: Nose, Eye, Nearknee, Nearfrontfetlock, Nearfrontfoot, Offknee, Offfrontfetlock, Offfrontfoot, Shoulder, Midshoulder, Elbow, Girth, Wither, Nearhindhock, Nearhindfetlock, Nearhindfoot, Hip, Stifle, Offhindhock, Offhindfetlock, Offhindfoot, Ischium.

We created 3 splits that contain 10 randomly selected training horses each (referred to as Horse-10). For each training set we took a subset of $5\%$ ($\approx 160$ frames), $10\%$ ($\approx 300$ frames), $20\%$ ($\approx 560$ frames), $50\%$ ($\approx 1470$ frames), and $90\%$ ($\approx 2580$ frames) of the frames for training, and then evaluated the performance on the training, test, and unseen ("out-of-domain") horses (i.e. the other horses that are in Horse-30, but were not in the given split of Horse-10). As metric we used mean average Euclidean error, which is computed by comparing the inferred poses for each body parts against the human prediction as well as percent correct key-point (PCK) values; i.e. what fraction of machine-applied points fall within a specific range of human-labeled ground-truth labels; although we use a matching threshold of $30\%$ of the head segment length (nose to eye for horse, which was computed by taking the median for all annotated images per horse) rather than $50\%$ as for MPII pose (Andriluka et al., 2014).

### 5.2 NETWORK VARIANTS

For this study we utilized a recently introduced an animal pose estimation toolbox called DeepLabCut (Mathis et al., 2018; Mathis & Warren, 2018; Nath et al., 2019). The TensorFlow-based network

architectures could be easily exchanged while keeping data loading, training, and evaluation consistent.

DeepLabCut (Mathis et al., 2018; Nath et al., 2019) is built on a subset of the deep feature detectors in DeeperCut (Insafutdinov et al., 2016). The feature detectors in DeepLabCut consist of residual networks (ResNets) (He et al., 2016) followed by deconvolutional layers to predict pose scoremaps and location refinement maps, which can then be used for predicting the pose while also proving a confidence score (Insafutdinov et al., 2016; Mathis et al., 2018). We utilize an output stride of 16 for the ResNets (achieved by atrous convolution) and then upsample the filter banks with deconvolutions by a factor of two to predict the heatmaps and location-refinement at 1/8th of the original image size scale. This gives a good balance of feature-map size and accuracy. However, the ratio can, of course, be changed and this affects speed, while still being relatively robust (Insafutdinov et al., 2016).

Here we also introduce a MobileNetV2 architecture that could be used within DeepLabCut (Sandler et al., 2018). MobileNetV2 utilizes depth-wise separable convolutions, inverted residual bottlenecks to significantly decrease the number of operations and memory needed while retaining high accuracy for ImageNet, object detection and image segmentation accuracy (Sandler et al., 2018). We configured the output-stride as 16 (by changing the (otherwise) last stride 2 convolution to stride 1). We utilized four variants of MobileNetV2 with different expansion ratios (0.35, 0.5, 0.75 and 1) as this ratio modulates the ImageNet accuracy from 60.3% to 71.8%, and pretrained models on ImageNet are available from TensorFlow (Abadi et al., 2016).

### 5.3 NETWORK TRAINING PARAMETERS

Most training parameters used here are consistent with the ones reported in DeepLabCut-2.0. (Nath et al., 2019). The training loss is defined as the cross entropy loss for the scoremaps and the location refinement error via a Huber loss with weight $0.05$; it is minimized via stochastic gradient descent with batch size 1 (Insafutdinov et al., 2016; Mathis et al., 2018). We use the following training schedule: $0.005$ for the first $10k$ iterations then $0.02$ onwards. For training from scratch, we had to start with a lower learning rate to avoid divergence of the loss and used $10^{-6}$ for the first $5,000$ steps, then $10^{-4}$ for the next $20k$, followed by $0.02$. We always trained for $100k$ iterations, unless noted. When training up to $600k$ iterations, we changed to $0.002$ after $430k$ iterations (as it is default for DeepLabCut).

*Speed Benchmarking*

Using an example from the horse dataset we evaluated one video with $11,178$ frames at resolutions $512 \times 512$, $256 \times 256$ and $128 \times 128$. We used batch sizes: $[1, 2, 4, 16, 32, 128, 256, 512]$, and ran all models for all 3 (training set shuffles) trained with $50\%$ of the data in a pseudo random order on an NVIDIA Titan RTX. For the benchmarking on a CPU we used shortened videos with merely $728$ frames; the CPU was an Intel Xeon CPU E5-2603 v4 @ 1.70GHz with 6 cores. We also changed the inference code from its numpy implementation (Mathis & Warren, 2018) to TensorFlow, which brings a $2 - 10\%$ gain in speed.

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

# A    APPENDIX

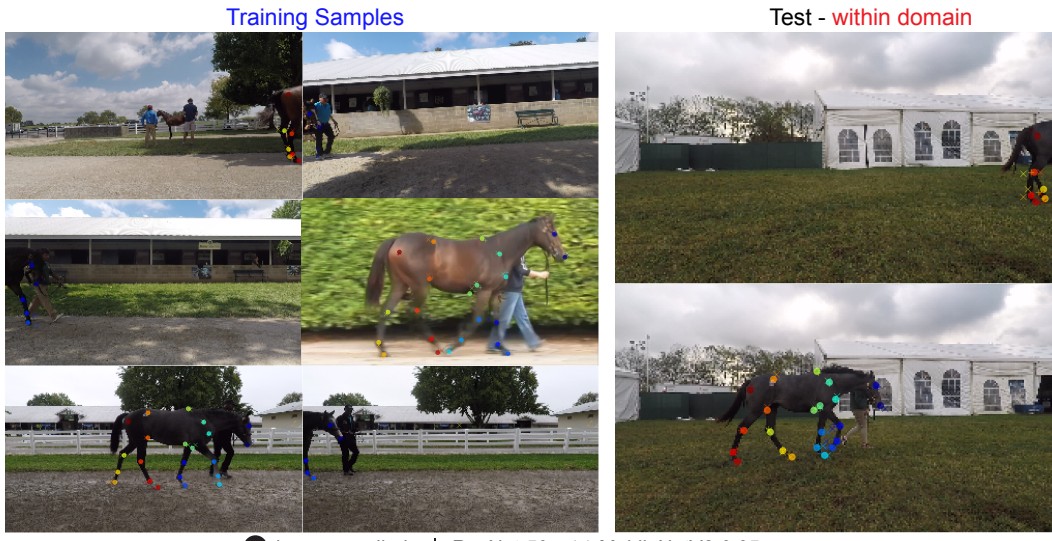

Figure 7: **Example frames with human and pretrained network annotations.** Here we show the smallest networks, namely ResNet-50 and the ultra-lightweight MobileNetV2-0.35, trained for $100,000$ iterations. Top Left set: example training images. Top Right: within domain test image results. Bottom: out-of-domain horses. Examples illustrate the challenges: varying coat colors, size changes, background, human legs, various postures, background horses, and partially occluded horses while they walk in and out of the video frames.

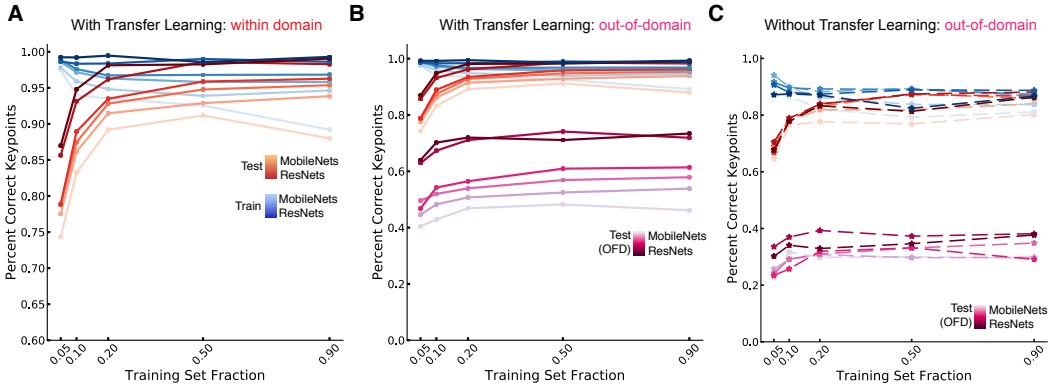

Figure 8: **Transfer Learning boosts accuracy (PCK). A:** Percent Correct Keypoint (PCK) vs. Training Set Fraction shows high performance for all networks on Horse-10. Darker to light red shades are test results for pretrained networks: ResNet-101, ResNet-50, MobileNetV2-1, MobileNetV2-0.75, MobileNetV2-0.5, MobileNetV2-0.35. Darker to lighter blue is for training, the same ordering as in test. All lines are averages of 3 splits (see Methods). **B:** Same as in A, plus the out-of-domain data (pink is for out-of-domain data on 20 unseen horses). **C:** Same as in B, but without pretraining on ImageNet.

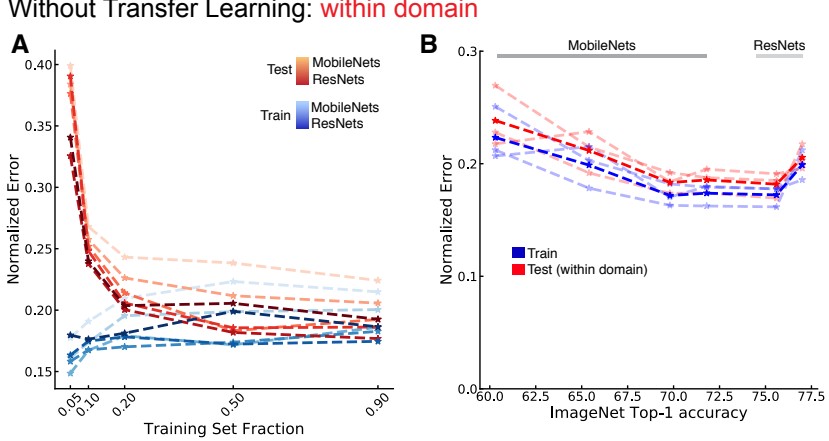

Figure 9: **Test and training performance when training from scratch.**) **A:** Normalized Error vs. Training Set Fraction of Horse-10. 5% is ≈ 160 frames. Darker to light red shades are test results for ResNet-101, ResNet-50, MobileNetV2-1, MobileNetV2-0.75, MobileNetV2-0.5, MobileNetV2-0.35. Darker to lighter blue is for training, same ordering as in test. **B:** Normalized Error vs. Network performance as ranked by the Top 1% accuracy on ImageNet, but here on Horse-10; namely, MobileNetV2-0.35, MobileNetV2-0.5, MobileNetV2-.75, MobileNetV2-1, ResNet-50, ResNet-101. Test data is in red, train is blue. This data is for 50% training set fraction.

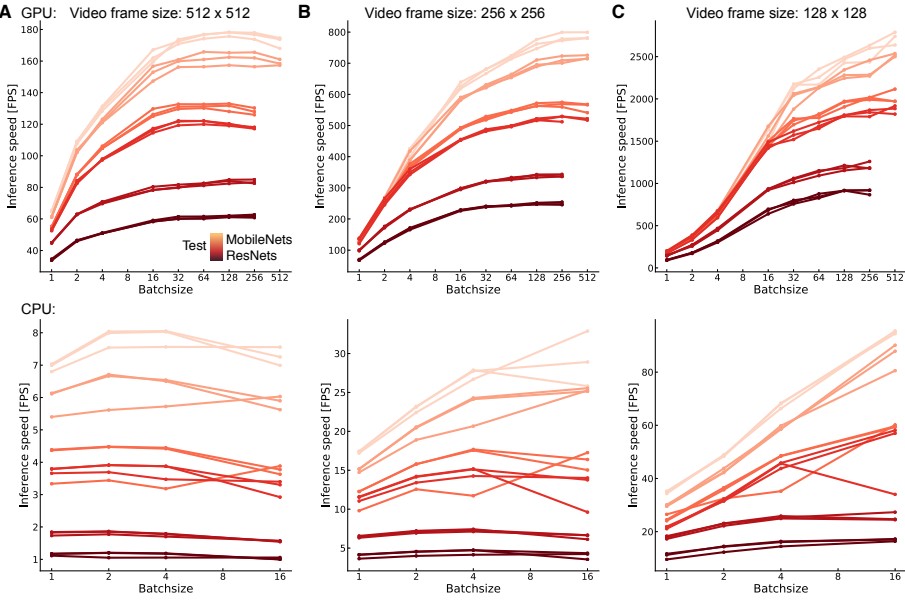

Figure 10: **Speed Benchmarking for ResNets and MobileNetV2s:** Inference speed for videos of different dimensions for all the architectures. **A-C:** FPS vs. batchsize, with video frame sizes as stated in the title. Three splits are shown for each network. MobileNetV2 gives a more than 2X speed improvement (over ResNet-50) for offline processing and about $40\%$ for batchsize=1 on a Titan RTX GPU. On CPU we found even larger gains.

Table 1: Batch size 1 (FPS): Mean inference speed for batchsize=1 and batchsize=256 (Table 2) for there different video frame sizes on a Titan RTX GPU. Video was $\approx 11,000$ frames long of a horse, with 22 bodyparts to be identified. See Methods for further details.

|                  | 128x128 | 256x256 | 512x512 |
|------------------|---------|---------|---------|
| MobileNetV2-0.35 | 195     | 132     | 65      |
| MobileNetV2-0.50 | 185     | 131     | 61      |
| MobileNetV2-0.75 | 185     | 129     | 55      |
| MobileNetV2-1    | 190     | 132     | 53      |
| ResNet-50        | 146     | 99      | 45      |
| ResNet-101       | 93      | 69      | 34      |

Table 2: Batch size 256 (FPS)

|                  | 128x128 | 256x256 | 512x512 |
|------------------|---------|---------|---------|
| MobileNetV2-0.35 | 2557    | 784     | 176     |
| MobileNetV2-0.50 | 2338    | 711     | 161     |
| MobileNetV2-0.75 | 2008    | 568     | 128     |
| MobileNetV2-1    | 1834    | 523     | 118     |
| ResNet-50        | 1208    | 339     | 84      |
| ResNet-101       | 902     | 249     | 62      |

