# OpenReview forum: "Pretraining boosts out-of-domain robustness for pose estimation"
_ICLR.cc/2020/Conference — Reject_

### Official Review · AnonReviewer2 · 2019-10-23
**Official Blind Review #2**

**Rating:** 1

**Review:**

The paper proposes a horse dataset to study transfer learning or so-called out-of-domain pose estimation. They also study which model is a better initialization model for pose estimation, and how to utilize transfer learning to get a better estimation model.

There are several questions from the paper:
1. why are they not using human pose estimation datasets, as there are already lots of them and that will be easier to compare with other models: MPII, COCO, AI challenge, CrowdPose, etc.  I am not fully convinced the horse dataset is better than human pose. In terms of out domain, authors can use pose pre-trained models to analysis horse pose prediction.
2. The analysis is good and with lots of experiments, however, the key part is that they do not provide a way to improve the overall performance for out-of-domain pose estimation.

**Experience Assessment:**

I have published one or two papers in this area.

**Review Assessment: Checking Correctness Of Derivations And Theory:**

I carefully checked the derivations and theory.

**Review Assessment: Checking Correctness Of Experiments:**

I carefully checked the experiments.

**Review Assessment: Thoroughness In Paper Reading:**

I read the paper at least twice and used my best judgement in assessing the paper.

---

> ### Author Response · Authors · 2019-11-06
> **Response to Reviewer #2**
>
> Review: The paper proposes a horse dataset to study transfer learning or so-called out-of-domain pose estimation. They also study which model is a better initialization model for pose estimation, and how to utilize transfer learning to get a better estimation model.
>
> There are several questions from the paper:
> 1. why are they not using human pose estimation datasets, as there are already lots of them and that will be easier to compare with other models: MPII, COCO, AI challenge, CrowdPose, etc.  I am not fully convinced the horse dataset is better than human pose. In terms of out domain, authors can use pose pre-trained models to analysis horse pose prediction.
>
>
> Firstly, typical human pose estimation benchmarks (MPII pose and COCO) contain many different individuals (>10k) in different contexts, but only few examples per individual (<10).
> This setting does not allow to ask how well an algorithm generalizes to different individuals that have slightly/strongly different appearance (and we do not feel it is wise/ethical to cluster individuals based on their appearance, etc to test this). Unlike the human pose estimation benchmarks, typically in animal pose estimation datasets that have many (annotated) poses per individual (>200) and only a few individuals (~10-100).
>
> We believe that the Horse Dataset we developed is a powerful dataset to study generalization across individuals and important for applications using small datasets. Furthermore using this dataset, we demonstrate that that “better” ImageNet networks transfer better as well as that task-driven training can match the performance of of transfer learning. However, we also found that for out-of-domain data, pretraining helps significantly, boosting performance up to 3 times (Fig 4). We think this is a novel result. It also shows that within domain/across domain horses are actually quite different.
>
> 2. The analysis is good and with lots of experiments, however, the key part is that they do not provide a way to improve the overall performance for out-of-domain pose estimation.
>
> One of our findings is that the out-of-domain test performance improves by 1.6% per percentage point of ImageNet performance (Fig 2B). Thus, our paper demonstrates that better (ImageNet pretrained) backbones will be more robust and contribute to close the gap between the within domain and out of domain performance.
>
> We sincerely thank you for your time and efforts, and hope this clarifies our contribution to the field! - The Authors

---

### Official Review · AnonReviewer1 · 2019-10-26
**Official Blind Review #1**

**Rating:** 1

**Review:**


= Summary
This paper analyzes the effect of ImageNet pretraining on out-of-domain visual recognition. Specifically, in this paper, the recognition problem is narrowed to pose estimation on a horse profile image dataset where the out-of-domain indicates horse IDs unseen during training. The paper presents a new horse pose estimation dataset and extensive experimental analysis to demonstrate the benefit of ImageNet pretraining.


= Decision
I would recommend to reject this submission mainly due to the shortcomings of the proposed dataset, which make the analysis and conclusion of the paper unconvincing.

First, the dataset is very limited in terms of diversity. It only contains 8K horse profiles images, each of which contains only a single horse, and only 30 horses of the same species appear in the images. Furthermore, since the dataset are sampled from video sequences, images of the same horse ID could be too similar in terms of appearance. Thus, all the images in the dataset seems within a single and very specific domain, "profiles of a small number of Thoroughbred horses", and not appropriate to evaluate "out-of-domain" robustness of pose estimation networks in consequence.

Second, the dataset split strategy is weird. Images of 10 horse IDs are considered as the "within domain" dataset while the others as "out-of-domain" dataset, and a subset of the within domain dataset is used for training or finetuning the pose estimation networks. This means that the networks could seriously overfitted to the small image set of 10 horse IDs. Thus, it is obvious that the networks will work very well with and without ImageNet pretraining on the within domain test set in which exactly the same 10 horses appear, and that they will not work well on the "out-of-domain" dataset due to the overfitting issue.

Also, focusing only on the "horse" class looks not proper. Note that ImageNet already contains many horse images, and ImageNet pretrained networks would have a capability to extract horse-related features. Thus, the advantage of ImageNet pretraining on the horse pose estimation task is not surprising but a result that many in this field can easily expect. If this is not a big issue, I rather would like to recommend to exploit existing human pose datasets (e.g., MPII) since they are larger enough in size, and guarantee a larger variety of poses and person appearances than the proposed horse dataset.


= Other comments
The manuscript is overall written clearly, but it is hard to understand the curves in the figures: colors are not clearly distinguishable (e.g., red vs. magenta), the roles of "MobileNets" and "ResNets" placed on top of upper horizontal bars are not unknown too (if they indicate that the top-2 accuracy scores are given by Resnets variants, why are the curves of MobileNets and ResNets connected?)



**Experience Assessment:**

I have read many papers in this area.

**Review Assessment: Checking Correctness Of Derivations And Theory:**

I assessed the sensibility of the derivations and theory.

**Review Assessment: Checking Correctness Of Experiments:**

I assessed the sensibility of the experiments.

**Review Assessment: Thoroughness In Paper Reading:**

I read the paper at least twice and used my best judgement in assessing the paper.

---

> ### Author Response · Authors · 2019-11-06
> **Response to Review #1: Part 1.**
>
> = Summary
> This paper analyzes the effect of ImageNet pretraining on out-of-domain visual recognition. Specifically, in this paper, the recognition problem is narrowed to pose estimation on a horse profile image dataset where the out-of-domain indicates horse IDs unseen during training. The paper presents a new horse pose estimation dataset and extensive experimental analysis to demonstrate the benefit of ImageNet pretraining.
>
> Thank you for your reviewing our work. Your comments essentially boil down to concerns with the novel Horse Dataset (“too small & little diversity”). This is disappointing, but here we hope to convince you that this dataset is important and can help us understand/improve pose estimation algorithms and representation learning more general.
>
> Firstly, typical human pose estimation benchmarks (MPII pose and COCO) contain many different individuals (>10k) in different contexts, but only few examples per individual (<10).
> In real world applications of pose estimation (beyond human pose estimation benchmarks) one naturally asks the following question: Assume you have an algorithm that performs pose estimation with very high accuracy (for biomechanics/neuroscience applications) on a given (individual) animal with all its poses (this is important to relate e.g. pose to coding in the brain). How well will it generalize to different individuals that have slightly/strongly different appearance? Unlike the human pose estimation benchmarks one deals with datasets that have many (annotated) poses per individual (>200) and only a few individuals (~10-100).
>
> We believe that the Horse Dataset we developed is a great dataset to study generalization across individuals and important for applications using small datasets. Furthermore using this dataset, we demonstrate that that “better” ImageNet networks transfer better as well as that task-driven training can match the performance of of transfer learning. However, we also found that for out-of-domain data, pretraining helps significantly, boosting performance up to 3 times (Fig 4). We think this is a novel result. It also shows that within domain/across domain horses are actually quite different.
>
> Below we added additional information that we hope will improve the score and highlight why these types of datasets and the results are important.
>
> = Decision
> I would recommend to reject this submission mainly due to the shortcomings of the proposed dataset, which make the analysis and conclusion of the paper unconvincing.
>
> We wish to take this opportunity to speak to why such new datasets are crucially important:
> (1) Animal pose estimation has important applications in neuroscience, ethology, technology, and medical studies (drug testing, etc. - see recent commentary in Nature: https://www.nature.com/articles/d41586-019-02942-5). The application of deep learning to animals is a budding field, empowered by the advances in human pose estimation, but has its own unique challenges - some of which we aim to address with this paper. Namely, small datasets, much much less than 8K are the norm - in fact, it is common for experimenters to only use 50-1000 images to create a training set, yet they demand good performance. Here are recent examples: Mathis et al 2018, Nath et al 2019, Graving et al 2019, Pereira et al 2019. Thus, we specifically wanted to create an animal benchmark dataset and test the relationship of how different architectures generalize across individuals. The data varies in horse color, the appearance of sunlight and shadow, and relative horse size as well as background. This makes the data set ideal for tests in robustness and generalization. If the term “out-of-domain” feels incorrect, we are very happy to revise this.
>
> (2) We tested the effect of pretraining on out-of-domain data on small datasets, which has not been done, to the best of our knowledge.
>
> (3) While the reviewer is absolutely correct that human pose benchmarks provide large datasets, they are not designed with an out-of-domain component (it has 40K individuals in 25K images). Nor do these benchmarks address (some of) the challenges that animal pose estimation researchers actually face (see above).
>
> [ continued below ]

---

> > ### Author Response · Authors · 2019-11-06
> > **Response to Reviewer #1: Part 2**
> >
> > Second, the dataset split strategy is weird. Images of 10 horse IDs are considered as the "within domain" dataset while the others as "out-of-domain" dataset, and a subset of the within domain dataset is used for training or finetuning the pose estimation networks. This means that the networks could seriously overfitted to the small image set of 10 horse IDs. Thus, it is obvious that the networks will work very well with and without ImageNet pretraining on the within domain test set in which exactly the same 10 horses appear, and that they will not work well on the "out-of-domain" dataset due to the overfitting issue.
> >
> > (1) We agree that models should work well on within domain data. That is what we find, but importantly, and this is one of our findings ImageNet accuracy predicts the within domain accuracy on pose estimation. This is consistent with what Kornblith et al. 2019 CVPR found this for object recognition.
> >
> > (2) Most importantly we find that better models generalize better to out-of domain horses. We do think that this result is surprising. Namely, better networks could also overfit more and thus generalize less. In fact, we find that the out-of-domain test performance improves by 1.6% per percentage point of ImageNet performance.
> >
> > (3) Now — going with your criticism — if we hypothesize that the “out of domain” and “within domain” horses are very similar. That would suggest:
> >
> >         a) the “generalization” to out of domain should be similar to within domain. But depending on the architecture the performance is up to 6 times worse (Fig 2B).
> >
> >         b) the slope ImageNet accuracy vs. Performance remains similar between within/out of domain. But again this is not the case. Instead, we find that better models are substantially better on out of domain data suggesting they the “understand the task” (Fig 2B).
> >
> >
> > Also, focusing only on the "horse" class looks not proper. Note that ImageNet already contains many horse images, and ImageNet pretrained networks would have a capability to extract horse-related features. Thus, the advantage of ImageNet pretraining on the horse pose estimation task is not surprising but a result that many in this field can easily expect. If this is not a big issue, I rather would like to recommend to exploit existing human pose datasets (e.g., MPII) since they are larger enough in size, and guarantee a larger variety of poses and person appearances than the proposed horse dataset.
> >
> > We believe this is a misunderstanding. We find that given enough training data, training from scratch, with purely task-driven training can match the performance of of transfer learning (Fig 5). Again this is something that has recently been highlighted in He’s et al. “Rethinking ImageNet pre-training” also for COCO keypoint detection. The Horse dataset allows us to go beyond their finding. Thus, while it has been previously shown that training from scratch can match performance on in-domain data for sufficiently large amount of training data and training times, we show it clearly cannot match performance of pretrained networks on out-of-domain data (Figure 5), especially on small datasets.
> >
> >
> > = Other comments
> > The manuscript is overall written clearly, but it is hard to understand the curves in the figures: colors are not clearly distinguishable (e.g., red vs. magenta), the roles of "MobileNets" and "ResNets" placed on top of upper horizontal bars are not unknown too (if they indicate that the top-2 accuracy scores are given by Resnets variants, why are the curves of MobileNets and ResNets connected?)
> >
> > Thank you for these constructive comments. We will update the red vs. magenta color scheme to be more clear. The MobileNets and ResNets bar/text is to guide the reader about which points on the graph are coming from the 4 MobileNet and 2 ResNet variants (order ranked by top 1% ImageNet accuracy). All the points are linked so one can appreciate the major findings of the paper, i.e. (1) networks that perform better on ImageNet perform better on pose estimation, both within and out-of-domain, and (2) pretraining absolutely matters on small datasets, something not tested elsewhere.
> >
> > We sincerely thank you for your time and efforts! - The Authors

---

### Public Comment · ~Yosuke_Shinya1 · 2019-09-29
**Related work**

Hi,

Would you clarify the difference from [1, 2] in the paper?
Thanks.

[1] Dan Hendrycks, Kimin Lee, Mantas Mazeika.
Using Pre-Training Can Improve Model Robustness and Uncertainty. ICML, 2019.
http://proceedings.mlr.press/v97/hendrycks19a.html
https://arxiv.org/abs/1901.09960
[2] Naman Jain, Sahil Shah, Abhishek Kumar, Arjun Jain.
On the Robustness of Human Pose Estimation. CVPRW, 2019.
http://openaccess.thecvf.com/content_CVPRW_2019/papers/Augmented%20Human%20Human-centric%20Understanding%20and%202D-3D%20Synthesis/Jain_On_the_Robustness_of_Human_Pose_Estimation_CVPRW_2019_paper.pdf
https://arxiv.org/abs/1908.06401

---

> ### Author Response · Authors · 2019-10-02
> **Clarifying related work**
>
> Dear Dr. Shinya, thanks for pointing us to the two papers [1,2].  Both of them—like our work—report on a beneficial effect of pre-training on robustness and we will definitely update our preprint to include a discussion of the relation to these two studies.
>
> We believe that our work complements and builds on these in two important ways: The key difference in our work is that we report on a different type of robustness that is of direct importance to a practical application we are trying to solve, namely accurate animal pose estimation under general conditions.
>
> In contrast, Jain et al [2] study robustness against adversarial attacks, and Hendrycks et al. [1] study CIFAR 10, CIFAR 100 and TinyImageNet instead of pose estimation. Hendrycks et al. [1] comes closer to our work than [2] as it shows a benefit for various types of robustness with out-of-domain detection being somewhat related to the problem that we are trying to solve. Note, however, that the analysis of out-of distribution-detection is quite different from the notion of out-of-domain robustness that we are testing.
>
> We believe the most relevant result from the two papers for our study is Table 4 in [1]:
> http://proceedings.mlr.press/v97/hendrycks19a/hendrycks19a.pdf (page 7)
>
> The performances in table 4 do not reflect task performance on out-of-domain test data but the ability to detect whether a sample is drawn from the original training distribution or from a corrupted version. In contrast, our work measures pose estimation performance on out-of-domain test data, for which we also have human annotated ground truth, and which is of high practical relevance.
>
> Again, thank you for the comment, and the pdf will be updated accordingly.

---

> > ### Public Comment · ~Yosuke_Shinya1 · 2019-10-03
> > **Re: Clarifying related work**
> >
> > Dear authors,
> >
> > Thank you for the very clear and kind explanation!

---

### Decision · Program_Chairs · 2019-12-19

**Decision:**

Reject

**Comment:**

The paper presents a new dataset, containing around 8k pictures of 30 horses in different poses. This is used to study the benefits of pretraining for in- and out-of-domain images.

The paper is somewhat lacking in novelty. Others have studied the same type of pre-training in the past using other datasets, which makes the dataset the main novelty. But reviewers raised many questions about the dataset, in particular about how many of the frames of the same horse might be similar, and of how few horses there are; few enough to potentially not make the results statistically meaningful. The authors replied to these questions more by appealing to standards in other fields than by explaining why this is a good choice. Apart from these crucial weaknesses, however, the research appears good.

This is a pretty clear reject based on lack of novelty and oddities with the dataset.